# Preparation and Evaluation of Release Formulation of γ-Oryzanol/Algae Oil Self-Emulsified with Alginate Beads

**DOI:** 10.3390/md17030156

**Published:** 2019-03-07

**Authors:** Kai-Min Yang, Po-Yuan Chiang

**Affiliations:** 1Department of Hospitality Management, MingDao Unicersity, 369 Wen-Hua Rd., ChangHua 52345, Taiwan; a9241128@gmail.com; 2Department of Food Science and Biotechnology, National Chung Hsing University, 250 Kuokuang Road, Taichung 40227, Taiwan

**Keywords:** γ-oryzanol, algae oil, self-emulsifying, alginate, controlled release

## Abstract

Self-emulsion improves solubility and bioavailability for γ-oryzanol/algae oil, and alginate beads can be used as controlled release carriers. In this study, self-emulsified alginate beads (SEABs) were prepared with different weight ratios of self-emulsion treatment (5%, 10%, 15%, 20%, and 30%) with alginate. We found that the microstructure with a surfactant of SEABs had a different appearance with alginate-based beads. The encapsulation of γ-oryzanol corresponded with the self-emulsion/alginate ratio, which was 98.93~60.20% with a different formulation of SEABs. During in vitro release, SEABs had the gastric protection of γ-oryzanol/algae oil, because γ-oryzanol and emulsion were not released in the simulated stomach fluid. When the SEABs were transferred to a simulation of the small intestine, they quickly began to swell and dissolve, releasing a higher content of the emulsion. We observed that the emulsion that formed had a bimodal distribution in the simulated intestinal fluid as a result of the hydrogel and emulsion droplets, leading to the formation of large aggregates. These results suggested that γ-oryzanol encapsulation within alginate beads via emulsification combined with gelation can serve as an effective controlled delivery system.

## 1. Introduction

γ-Oryzanol was first isolated in 1954 by Kaneko and Tsuchiya from the unsaponifiable fraction of rice bran as a crystalline substance; it is mainly composed of esters of trans-ferulic acid with phytosterols [1,2]. Among these, cycloartenol, β-sitosterol, 24-methylenecycloartenol, and campesterol predominate. Since then, γ-oryzanol has allowed for the identification of more than 20 ferulic acid esters of triterpene alcohols and sterols. For industrial procedures, soapstock is the richest source of γ-oryzanol, with the refining process used to prepare soapstock involving a deacidification treatment of rice bran oil [3]. γ-Oryzanol has received considerable attention due to its numerous positive qualities, including the fact that it is anticarcinogenic, anti-inflammatory, antihyperlipidemic, antidiabetic, and neuroprotective [1,2,4]. Previous studies have shown that γ-oryzanol is safe in terms of subacute toxicity, chronic toxicity, teratogenicity, and developmental toxicity [4,5]. DHA was a popular nutritional with the consumer market in view of the clear evidence of the health benefits, including development of the eyes and brain, antihyperlipidemic, antihypertensive, anti-inflammatory, and antiarrhythmic effects [6]. The rate of DHA biosynthesis is low and insufficient to meet the physiological demands of humans, of which dietary intake is the primary source. Algae oil is a commercially successful alternative via microalgae fermentation, which has high amounts of DHA, shortens growth, and has good sensory and safety profiles [6,7].

According to the Biopharmaceutical Classification System (BCS), the bioavailability of pharmaceuticals delivered orally is influenced by their solubility and intestinal permeability. The BCSII includes low-solubility groups that cause drug absorption via the gastrointestinal tract to be the limiting factor in terms of bioavailability, which can be useful in achieving suitable blood levels [8,9]. Emulsion formulations have the potential to enhance the solubilization of BCSII drugs. The solubility and bioavailability of BCSII drugs can be improved with certain kinds of delivery systems, such as those involving microencapsulation, nanoemulsion, solid lipid nanoparticles, and self-emulsifying drug-delivery systems (SEDDS). Self-emulsifying drug-delivery systems are composed of a given drug, oils, surfactants, and sometimes co-solvents. Self-emulsifying drug-delivery systems consist of emulsion concentrates, but they are not themselves emulsions [8]. However, when they are subjected to mild agitation in the aqueous environment of the stomach, they easily form stable emulsions. Self-emulsifying drug-delivery systems have been marketed in the form of liquid or semisolid products, but these forms have a few shortcomings, especially in terms of the manufacturing process. These shortcomings have led to the development of SEDDS in solid-dosage forms, which besides offering improved solubility, also offer further advantages over liquid forms, such as minimizing gastrointestinal tract irritation without lowering drug bioavailability [8,10]. In the market, SEDDS formulations of Tretinoin, Cyclosporin A, and Saquinavir have already been introduced as commercial products [9]. Emulsification combined with gelation has been suggested as an alternative means of encapsulating several compounds, including sensitive or hydrophobic biologicals. Together, emulsification and gelation provide an inert environment, which allows the embedded drugs to maintain greater biological activity and to have strong stability for the duration of their shelf life [9,11,12].

Ferulate, one of the functional groups in γ-oryzanol, provide antioxidative capacity, and enhance the stabilization of algae oil. Simultaneously, γ-oryzanol and algae oil lead synergistic effects on cardiovascular disease. The application of γ-oryzanol in medical and functional food systems may be limited by its low water solubility, poor bioavailability, and rapid metabolism. In this study, we designed a SEDDS for γ-oryzanol/algae oil, with alginate beads used as carriers. We then evaluated the release characteristics in the gastrointestinal tract of the γ-oryzanol/algae oil contained in the self-emulsified alginate beads (SEABs). More specifically, we measured its particle size, turbidity, γ-oryzanol amount, and antioxidant capacity.

## 2. Results and Discussion

### 2.1. Physical Characteristics

Oil was one of the essential excipients in the SEDDS formulation, not only because it can solubilize marked amounts of γ-oryzanol and facilitate self-emulsification, but also, and primarily, because it can increase the fraction of γ-oryzanol transported via the intestinal lymphatic system, thereby increasing its absorption from the gastrointestinal tract [8,9]. In this study, the γ-oryzanol containing SEABs comprised 60% algal oil (which contained 40 mg of γ-oryzanol per mL), 24% Tween 80, and 16% Span 80. We used algal oil because it provides the most “natural and functional” basis for excipients. In our previous studies, the n-3 PUFA of algal oil was 30.3 g/100 g, and they have excellent oxidative stability (Ea: 96.8 kj/mol) [7]. The Tween and Span used as non-ionic surfactants are generally recognized as safe, and the acceptable daily intake set by WHO is 25 mg/kg of body weight [8]. In addition, they are more compatible with biological systems and less affected by pH and ionic strength. γ-Oryzanol/algae oil self-emulsion was assessed using the self-emulsification test, the results of which indicated that the spontaneity, homogeneity, and dispersibility of the self-emulsion were good. The appearance of the emulsion formulation was milky, while the z-average and polydispersity index (PDI) of the droplets were 149 nm and 0.246 (data not shown).

Alginates are naturally occurring substances that can be considered a form of block polymer, which mainly consist of mannuronic acid, guluronic acid, and mannuronic–guluronic blocks. The dropwise addition of aqueous alginate solution to an aqueous solution containing calcium ions and/or other di- and polyvalent cations causes the formation of gel in a spherical shape, termed “an alginate bead,” and such beads have been widely used as drug carriers for oral administration [12,13]. The alginate bead formation of 5~10% SEABs with varying ratios of γ-oryzanol self-emulsion resulted in spherical shapes with a narrow size distribution. When a higher load of γ-oryzanol/algae oil self-emulsion (>15%) was used, the tail became more distinct because of extrusion dripping (Figure 1). The results resembled those reported by other studies, where varying ratios of hydrogel to oil resulted in spherical, pear-shaped beads by reductions of interfacial tension, whereas tear-shaped beads, as well as spherical beads, formed with the lowest surface/volume ratio that were beneficial to the diffused compounds [14].

The encapsulation, expressed as the percentage of encapsulated γ-oryzanol/algae oil relative to the self-emulsion used, decreased as alginate concentration increased. We found that γ-oryzanol/algae oil encapsulation levels of 5%, 10%, 15%, 20%, and 30% SEABs were 98.93%, 73.04%, 68.73%, 66.73%, and 60.20%, respectively (Table 1). Hydrocolloid was found to improve emulsion stability. This may be attributable to an increase in the viscosity of the continuous phase surrounding the oil droplets, thus restricting their movement, or the adsorption/precipitation of the gum in the oil–water interphase causing a reduction in interfacial tension [14,15].

Some studies have indicated that alginate-based beads are characterized by a porous and collapsed structure, and a clear ridge structure indicates strong gelation [16,17]. The use of calcium chloride with alginate leads to the cross-linking and aggregation of alginate, where the exchange of divalent ions of calcium during the reticulation process is responsible for the creation of a strong network, leading to a very strong matrix [18]. Figure 2d shows insoluble calcium salt in a calcium/alginate monomer ratio to ensure strong bead formation. When hydrogel encapsulates a hydrophobic substance, tiny grains can be found on the surface of the resulting particles. However, if a surfactant is also used, the grains have a smoother surface. These characteristics are caused by the sublimation of water crystals from the freeze-dried alginate matrix, resulting in void spaces and minor structural shrinkage. The difference in the treatment of surfactant shown in Figure 2 is that the bead has cracks. The other figure images show that the varying forms of SEABs had holes in them because of the plasticizing effect of the surfactant [19].

### 2.2. Release Characteristics

The potential for the controlled release of drugs from polymers has received considerable attention because controlled release could make it possible to retain the optimal concentration of a drug at a desired location in the body [12,14,20]. The ideal oral delivery system should overcome acid-base hydrolysis and enzyme degradation in the gastrointestinal tract, providing local treatment by delivering the drug as closely as possible to the target site and thereby reducing the incidence of systemic side effects.

The pharmacological activities of γ-oryzanol correspond to its antioxidant activity [21]. Due to the presence of the phenolic acids in its composition, its anti-ulcer and anti-inflammatory intestinal effects are a matter of great interest. In this study, we analyzed the γ-oryzanol release and 1,1-diphenyl-2-picrylhydrazyl (DPPH) scavenging of the SEAB when it was placed in simulated stomach fluid and found that the various forms of SEABs provide similar levels of protection to γ-oryzanol, given that the differences in γ-oryzanol release are not significant (*p* < 0.05). However, there were significant differences in the DPPH scavenging exhibited by the various formulations of SEABs (Figure 3). Initially, the DPPH scavenging levels of the 5%, 10%, 15%, 20%, and 30% SEABs were 9.7%, 15.1%, 26.0%, 20.7%, and 35.81%, respectively. When placed in the simulated stomach fluid, the DPPH scavenging levels increased, reaching their highest levels at 80 min, with those levels being 33.0%, 35.9%, 40.1%, 41.9%, and 47.7%, respectively, and then decreasing thereafter (Figure 3). Increasing the surfactant concentration had little impact on the antioxidant activity of the SEAB, a finding that can be attributed to the effects of the γ-oryzanol [22]. We found that the SEAB, the protection of the γ-oryzanol amount, was not consistent with DPPH scavenging. We expect that this is a reversible change of γ-oryzanol, attributable to the constitution of simulated stomach fluid. Simultaneously, we observed that in the simulated gastric fluid, γ-oryzanol was not detected and the emulsion was formed (data not shown).

When placed in simulated intestinal fluid, each form of the SEAB exhibited signs of swelling and erosion. Meanwhile, the aqueous environment produced a reconstituted emulsion of the γ-oryzanol/algae oil self-emulsion. The γ-oryzanol was solubilized in the oily core and on the interface of the emulsion structures [23,24]. The turbidity of the intestinal fluid corresponded with the emulsion formed from the SEAB. At 50 min after being placed in the simulated intestinal fluid, the 5%, 10%, 15%, 20%, and 30% SEABs caused turbidity levels of 14.8%, 14.3%, 22.5%, 40.3%, and 62.0%, respectively. At 100 min, those levels were 33.2%, 37.1%, 51.2%, 71.7%, and 90.15%, respectively. At 125 min, they were 50.1%, 61.9%, 71.2%, 90.4%, and 98.1%, respectively. We found that the release of 5% and 10% SEABs at 100 min was lower than 50% because of the strong network leading to strong gelation (Figure 4). With the 5~30% SEABs, we observed the droplet size of the emulsion in the simulated intestinal fluid. These particles exhibited a bimodal distribution, with a population of small droplets of approximately 129~215 nm in size and a population of large droplets of approximately 750~1916 nm in size (Table 2). At sufficiently low hydrogel concentrations, agglutination between oppositely charged biopolymers and emulsion droplets occurred, leading to the formation of large aggregates. When fluid secretion and peristalsis occurred, the large droplets in the simulated intestinal fluid were transformed into small droplets, the γ-oryzanol had to be hydrolyzed enzymatically from the large droplets, and its emulsification enhanced its enzyme affinity [25].

The pH responsivity of alginates is high because alginates are molecular polymers that undergo volume or phase transition when the pH value of the external environment changes. As the test results of the SEABs indicated (Figure 5), the first phase might result from the negligible dissociation in simulated gastric fluid during the first two hours. In the second phase, meanwhile, we observed a burst-like release pattern, because Tween is a non-ionic substance that has a relatively weak interaction with anionic biopolymer molecules [26].

## 3. Materials and Methods

### 3.1. Materials

Food-grade sodium alginate (low viscosity; 160–200 mPa of 2% solution) and calcium chloride were purchased from Gemfont Corporation (Taipei, Taiwan). γ-Oryzanol and DPPH were purchased from Sigma Chemical Co. (St. Louis, MO, USA). Tween 80, Span 80, and other analytical grade chemicals used in this study were purchased from Chemical Co., Ltd. (Miaoli, Taiwan).

### 3.2. Preparation of γ-Oryzanol SEAB

To prepare the γ-oryzanol/algae oil self-emulsion, γ-oryzanol (2.4 g) was first added to algal oil (60 mL), and the mixture was then stirred (400 rpm) until clear. Next, Tween 80 (24 g) and Span 80 (16 g) were added to the mixture. The γ-oryzanol SEABs were entrapped in calcium alginate (Ca–alginate) beads through ionotropic gelation in various compositions, as shown in Table 1. Specifically, 50 mL of alginate and γ-oryzanol self-emulsion were extruded through a coaxial bead generator (Unit-Varj1, Nisco Engineering AG, Switzerland) and then dripped into a calcium chloride gelling solution to form the γ-oryzanol self-emulsion, Ca–alginate beads. The tip of a needle was fixed at 10 cm above the surface of the gelling bath. The gelling solution was gently stirred with a magnetic stirrer (100 rpm) to prevent the beads from sticking together. After 30 min of bead formation in the gelling bath, the beads were collected by filtration and rinsed sequentially with distilled water and 95% ethanol.

### 3.3. Encapsulation Efficiency

In the next step, 200 mg of SEABs were ground in a mortar and dispersed with isopropanol (10 mL) in a volumetric flask, and then stirred at 500 rpm for 1 h. The solution was then filtered through a 0.45-mm filter. The γ-oryzanol content was determined using UV-visible spectrophotometry at 327 nm [27]. The drug-entrapment efficiency was determined using the following formula:

Encapsulation efficiency (%) = γ-oryzanol in low-gelation alginate/γ-oryzanol in SEAB × 100



### 3.4. Scanning Electron Microscopy

The microstructures of the SEABs were observed with an SEM (Model ABT-150S, Topcon Corp., Tokyo, Japan). The samples were placed on a double-sided adhesive tape fixed on an aluminum stub. The samples were then covered with a gold-palladium coating (Model JBS-ES 150, Ion Sputter Coater, Topcon Corp., Tokyo, Japan). The accelerating potential was 15 kV.

### 3.5. In Vitro Release

The SEABs were evaluated for in vitro drug release in simulated gastrointestinal fluids in accordance with previous research [14]. Simulated gastric fluid was prepared by adding 2 g NaCl and 6 mL 12 N HCl to 500 mL of distilled water, adjusting the pH to 1.2 by adding 0.01 M HCl, and then adding distilled water up to 1 L. The simulated intestinal fluid consisted of 0.2 M, pH 6.8 phosphate buffer. The various forms of SEABs (0.2 g) were suspended in 100 mL of simulated gastric fluid for 2 h, and then transferred to 100 mL of simulated intestinal fluid for 4 h. The various forms of SEABs were placed in simulated gastric fluid (or simulated intestinal fluid) at a temperature of 37 °C and stirred with a paddle at under 50 rpm. In the simulated gastric fluid phase, we analyzed the γ-oryzanol content, DPPH scavenging, and turbidity at regular time intervals. In the simulated intestinal fluid phase, we analyzed the turbidity and droplet size.

#### 3.5.1. Analysis of γ-Oryzanol Content

The various forms of SEABs were extracted with isopropanol, and the γ-oryzanol content was determined by UV-visible spectrophotometry at 327 nm.

#### 3.5.2. DPPH Scavenging

The DPPH radical-scavenging activity was detected using the same methods as in previous research [27]. The various forms of SEABs were extracted with isopropanol dissolved in an ethanol solution (0.2 mM) containing DPPH radicals. After shaking and incubation for 30 min, the sample was measured for UV absorbance at 517 nm. The absorbance of a sample (As), a control (where the sample was replaced with distilled water, Ac), and a blank (Ab) were measured by a spectrophotometer at 517 nm. The DPPH-scavenging activity was calculated with the following equation:
DPPH − scavenging activity (%) = 1 − (As − Ab)/Ac × 100



#### 3.5.3. Turbidity Measurements

The SEABs were gently separated from the simulated intestinal fluid using the same methods as in previous research [28]. Then, the solution was passed through a 0.45-mm filter, and the permeate was collected. The turbidity of the aqueous permeate samples was measured at 600 nm.

#### 3.5.4. Droplet Size Measurements

Dynamic light scattering was used to determine the droplet sizes of the emulsions (Zetasizer Nano-ZS, Malvern, UK). The samples were diluted to a droplet concentration of approximately 1/10, with an appropriate buffer to prevent multiple scattering effects. The foundation of this technique is based on the scattering of light by moving particles due to Brownian motion in a liquid. The movement of the particles was then related to the size of the particles. Each recorded measurement was an average of eight runs. All the samples were measured at least in duplicate at 25 °C. The instrument reports the mean particle diameter (z-average), PDI range, and size distribution.

### 3.6. Statistical Analysis

All of the experiments were performed in triplicate, and all the data were expressed in the form of mean ± standard deviation of the mean. Analysis of variance was conducted with SPSS 10.0 (SPSS, Chicago, IL, USA) to analyze the data obtained for the same group. Variances were analyzed using the Statistical Analysis System (2000) software (SAS Inst., Inc., Cary, NC, USA). To test the significance of the differences between paired means, Duncan’s multiple range test was used. A confidence level of *p* < 0.05 was applied to judge the significance of each difference.

## 4. Conclusions

The experimental results indicated that the γ-oryzanol/algae oil SEABs were prepared successfully through emulsification combined with gelation, which improved solubility for γ-oryzanol/algae oil and the liquid transformed into a solid. The pH responsivity of alginate prevented the release of γ-oryzanol/algae oil in the upper gastrointestinal tract, while subsequently allowing the emulsion formulation and release of γ-oryzanol upon the arrival of the beads in the intestinal fluid. The controlled release of SEABs was influenced by cross-linking them with alginate and emulsification treatment. Therefore, it can be suggested that SEABs constitute an effective delivery system for the oral administration of γ-oryzanol/algae oil. In the future, the SEAB production process and formulation parameters can be easily and successfully applied to various biologicals with a wide range of medical and pharmaceutical applications.

## Figures and Tables

**Figure 1 marinedrugs-17-00156-f001:**
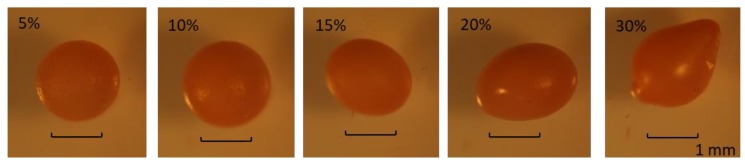
Photographic images of γ-oryzanol/algae oil self-emulsified alginate beads (SEABs) with different formulations.

**Figure 2 marinedrugs-17-00156-f002:**
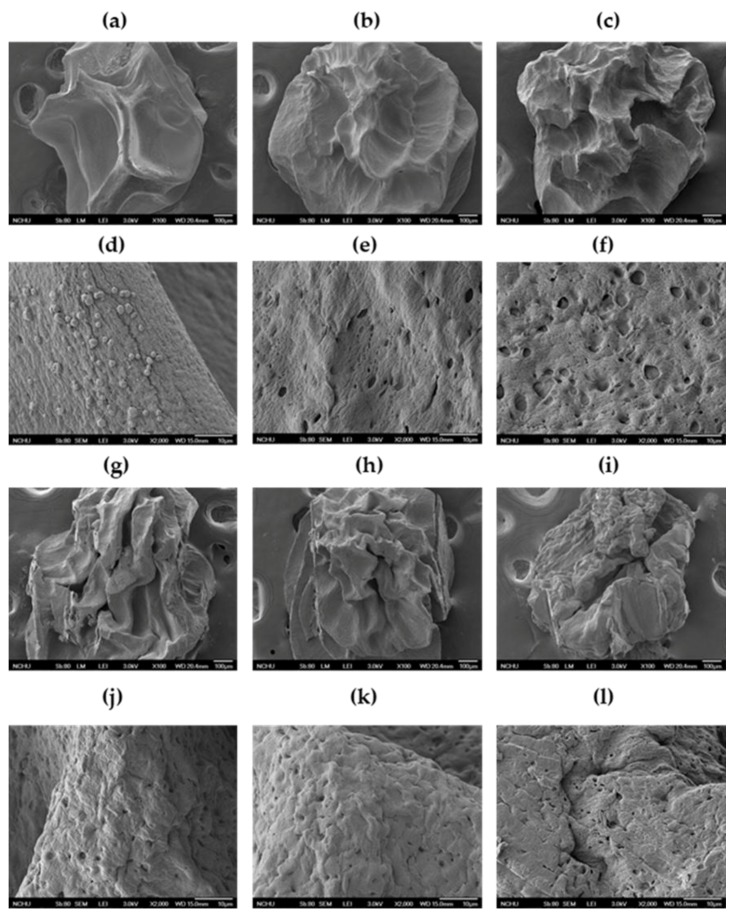
SEM micrographs of SEABs in (**a**) non-self-emulsified, (**b**) 5% SEAB, (**c**) 10% SEAB, (**g**) 15% SEAB, (**h**) 20% SEAB, and (**i**) 30% SEAB forms, at 100× magnification. SEM micrographs of SEABs in (**d**) non-self-emulsified, (**e**) 5% SEAB, (**f**) 10% SEAB, (**j**) 15% SEAB, (**k**) 20% SEAB, and (**l**) 30% SEAB forms, at 5000× magnification.

**Figure 3 marinedrugs-17-00156-f003:**
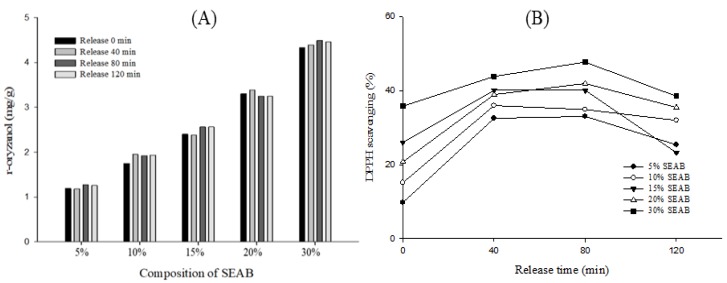
The release of γ- oryzanol/algae oil SEABs in simulated stomach fluid: (**A**) γ-oryzanol and (**B**) 1,1-diphenyl-2-picrylhydrazyl (DPPH) scavenging.

**Figure 4 marinedrugs-17-00156-f004:**
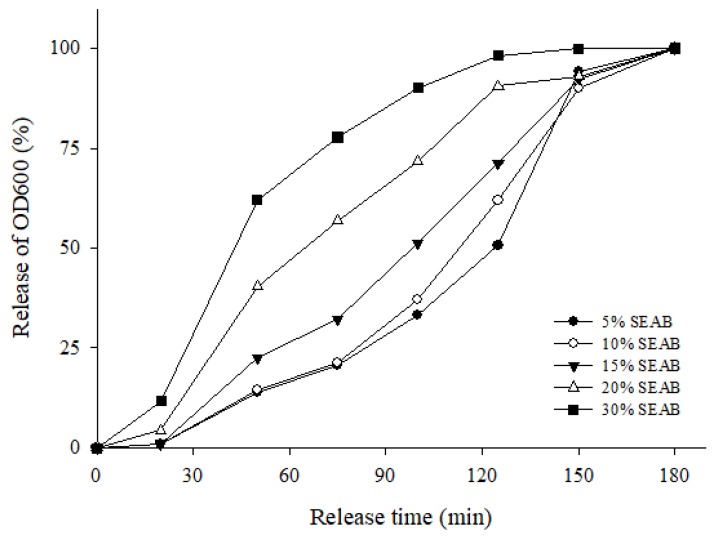
The release of γ- oryzanol/algae oil SEABs in the simulated intestinal fluid.

**Figure 5 marinedrugs-17-00156-f005:**
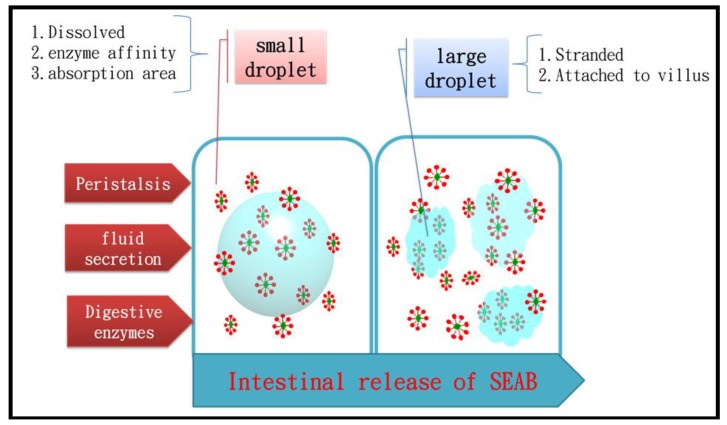
Graphical abstract of γ-oryzanol SEAB release in simulated intestinal fluid.

**Table 1 marinedrugs-17-00156-t001:** The compositions of γ-oryzanol/algae oil SEABs formulations.

	γ-Oryzanol Self-Emulsified Alginate Beads				
	5%	10%	15%	20%	30%
**Composition (%)**					
algae oil *	3.0	6.0	9.0	12.0	18.0
Tween 80	1.2	2.4	3.6	4.8	7.2
Span 80	0.8	1.6	2.4	3.2	4.8
alginate *	95.0	90.0	85.0	80.0	70.0
**Encapsulation (%)**					
γ-oryzanol	98.93	73.04	68.75	66.73	60.20

Data presented are in mean ± SD (*n* = 3), with each letter indicating significant variations at *p* < 0.05. * Algae oil contained 40 mg of γ-oryzanol per mL; 2.5% alginate aqueous solution.

**Table 2 marinedrugs-17-00156-t002:** The droplet size distribution of SEABs in the simulated intestinal fluid.

	γ-Oryzanol/Algae Oil SEABs				
	5%	10%	15%	20%	30%
Droplet Size in 50 min					
small droplet	142	215	163	161	163
large droplet	750	1434	1434	1573	1573
Droplet Size in 100 min					
small droplet	155	205	163	129	196
large droplet	832	1369	1434	1725	1725
Droplet Size in 125 min					
small droplet	129	171	181	171	196
large droplet	823	1916	1892	1807	1782

Data presented are in mean ± SD (*n* = 8).

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
