# Peer review of "Preparation and Evaluation of Release Formulation of γ-Oryzanol/Algae Oil Self-Emulsified with Alginate Beads"

_marinedrugs, 2019, doi:10.3390/md17030156_

Round 1

Reviewer 1 Report

I've read with attention the paper entitled "Preparation and evaluation of release formulation of 3 γ-oryzanol/algae oil self-emulsified with alginate 4 beads" that is potentially of interest. The methodology applied seems to be valid and the obtained results are reliable and adequately discussed. My only suggestion is to more clearly specify the research perspectives on the short term of the researcher discoveries

Author Response

Dear Editor

Thank you for considering the revised version of our manuscript: marinedrugs-457126, entitled " Preparation and evaluation of release formulation of γ-oryzanol/algae oil self-emulsified with alginate beads", by Yang. et al. for publication in Marine Drugs. We are thankful to the referees and the Editor for pointing out some important modifications needed in the report. We believe that the comments have been highly constructive and very useful to restructure the manuscript. We have thoughtfully taken into account these comments. The explanation of what we have changed in response to the reviewers’ concerns is given point by point in the following pages. The manuscript has been revised as suggested, and using the "Track Changes" function in Microsoft Word.

1. Academic Editor's Comments:

the language need major improvement.

The authors’ Answer:

I have completed the English editing and attached a certificate.

Many typographical errors and abbreviations have been revised. All the lines and pages indicated above are in the revised manuscript. We hope that all these changes fulfill the requirements to make the manuscript acceptable for publication in Marine Drugs.

Looking forward to hearing from you soon.

Sincerely yours,

Kai-Min Yang

Reviewer 2 Report

The manuscript entitled “ Preparation and evaluation of release formulation of γ-oryzanol/algae oil self-emulsified with alginate beads” by Yang KM et al. describes the effect of alginate beads emulsification on γ-oryzanol/algae oil. Please see comments below:

1) As per line 65, how does γ-oryzanol helps oxidation stability, Please elaborate.

2) In line 97, why does 15-30% formulation of  γ-oryzanol/algae oil SEABs shows tail like shapes?

3) In line 101, What are the key interactions or forces involved in encapsulation with alginate beads. Provide evidence.

4) In fig 3b, Why DPPH scavenging efficiency decreased at 120 min release time, provide an explanation. What if prolonged the release time until 180 min.

5) In line 155, Did the authors run γ-oryzanol only as a control to compare the effect of SEABs?

6) In Fig 5, When large droplet splits into small droplets, what exactly the difference observed when compared the effect of small droplet?

7) In line 260, how does the authors define cross-linking term.

8) References need to be according the journal format and some of the references does not have complete list of page numbers. Please make the references consistent across the text or references.

Author Response

Dear Editor

Thank you for considering the revised version of our manuscript: marinedrugs-457126, entitled " Preparation and evaluation of release formulation of γ-oryzanol/algae oil self-emulsified with alginate beads", by Yang. et al. for publication in Marine Drugs. We are thankful to the referees and the Editor for pointing out some important modifications needed in the report. We believe that the comments have been highly constructive and very useful to restructure the manuscript. We have thoughtfully taken into account these comments. The explanation of what we have changed in response to the reviewers’ concerns is given point by point in the following pages. The manuscript has been revised as suggested, and using the "Track Changes" function in Microsoft Word.

1. Academic Editor's Comments:

the language need major improvement.

The authors’ Answer:

I have completed the English editing and attached a certificate.

2. Academic Editor's Comments:

2.1 As per line 65, how does γ-oryzanol helps oxidation stability, Please elaborate.

The authors’ Answer:

I'm sorry about that I forgot to provide sufficient information, I have reconfirmed and the revised portions. Please see line 67-69.

3. Academic Editor's Comments:

3.1 In line 97, why does 15-30% formulation of γ-oryzanol/algae oil SEABs shows tail like shapes?

The authors’ Answer:

I'm sorry about that I forgot to provide sufficient information, I have reconfirmed and the revised portions. Please see line 99-102.

4. Academic Editor's Comments:

4.1 In line 101, What are the key interactions or forces involved in encapsulation with alginate beads. Provide evidence.

4.2 In line 260, how does the authors define cross-linking term.

The authors’ Answer:

Thank for your comments, which have presented line 92-96 and line 117-120 in manuscript.

5. Academic Editor's Comments:

In fig 3b, Why DPPH scavenging efficiency decreased at 120 min release time, provide an explanation. What if prolonged the release time until 180 min.

The authors’ Answer:

5.1 The change in antioxidant capacity to a in this study is worthy of future research. We think it may be caused by simulated gastric fluid interference, but it is impossible to confirm whether this result is a reversible reaction

5.2 The most of the orally administered drugs display region-specific absorption that could be related to differential drug solubility and stability in different regions of the GI tract as a result of changes in environmental pH, degradation by enzymes present in the lumen of the intestine or interaction with endogenous compounds. We refer to other studies with 120 points as the simulated stomach.

6. Academic Editor's Comments:

In line 155, Did the authors run γ-oryzanol only as a control to compare the effect of SEABs?

The authors’ Answer:

During the experiment, fatty acid analysis of SEAB by GC, showing interference with Tween 80 and Span 80.

7. Academic Editor's Comments:

In Fig 5, When large droplet splits into small droplets, what exactly the difference observed when compared the effect of small droplet?

The authors’ Answer:

7.1 Droplet size is an important parameter in the development of self-emulsifying drug-systems as it influences the rate of drug release as well as their in vivo stability. Smaller droplets have a greater surface area, which subsequently increases the rate of drug release.

7.2 Thank for your comments, which have presented line185-188 and line 268-270 in manuscript.

Many typographical errors and abbreviations have been revised. All the lines and pages indicated above are in the revised manuscript. We hope that all these changes fulfill the requirements to make the manuscript acceptable for publication in Marine Drugs.

Looking forward to hearing from you soon.

Sincerely yours,

Kai-Min Yang
